# Predictive Potential of C_max_ Bioequivalence in Pilot Bioavailability/Bioequivalence Studies, through the Alternative ƒ_2_ Similarity Factor Method

**DOI:** 10.3390/pharmaceutics15102498

**Published:** 2023-10-20

**Authors:** Sara Carolina Henriques, Paulo Paixão, Luis Almeida, Nuno Elvas Silva

**Affiliations:** 1Research Institute for Medicines (iMed.ULisboa), Faculty of Pharmacy, Universidade de Lisboa, 1649-003 Lisboa, Portugal; ppaixao@ff.ulisboa.pt; 2BlueClinical Ltd., Senhora da Hora, 4460-439 Matosinhos, Portugal; lalmeida@blueclinical.pt

**Keywords:** bioequivalence, generic medicinal products, pilot studies, ƒ_2_ factor, pharmacokinetics, modelling and simulation, pharmacokinetic simulation

## Abstract

Pilot bioavailability/bioequivalence (BA/BE) studies are downsized trials that can be conducted prior to the definitive pivotal trial. In these trials, 12 to 18 subjects are usually enrolled, although, in principle, a sample size is not formally calculated. In a previous work, authors recommended the use of an alternative approach to the average bioequivalence methodology to evaluate pilot studies’ data, using the geometric mean (G_mean_) ƒ_2_ factor with a cut off of 35, which has shown to be an appropriate method to assess the potential bioequivalence for the maximum observed concentration (C_max_) metric under the assumptions of a true Test-to-Reference Geometric Mean Ratio (GMR) of 100% and an inter-occasion variability (IOV) in the range of 10% to 45%. In this work, the authors evaluated the proposed ƒ_2_ factor in comparison with the standard average bioequivalence in more extreme scenarios, using a true GMR of 90% or 111% for truly bioequivalent formulations, and 80% or 125% for truly bioinequivalent formulations, in order to better derive conclusions on the potential of this analysis method. Several scenarios of pilot BA/BE crossover studies were simulated through population pharmacokinetic modelling, accounting for different IOV levels. A redefined decision tree is proposed, suggesting a fixed sample size of 20 subjects for pilot studies in the case of intra-subject coefficient of variation (ISCV%) > 20% or unknown variability, and suggesting the assessment of study results through the average bioequivalence analysis, and additionally through G_mean_ ƒ_2_ factor method in the case of the 90% confidence interval (CI) for GMR is outside the regulatory acceptance bioequivalence interval of [80.00–125.00]%. Using this alternative approach, the certainty levels to proceed with pivotal studies, depending on G_mean_ ƒ_2_ values and variability scenarios tested (20–60% IOV), were assessed, which is expected to be helpful in terms of the decision to proceed with pivotal bioequivalence studies.

## 1. Introduction

The approval of brand-name and generic drugs under the European Medicines Agency (EMA) [1] and US Food and Drug Administration (FDA) [2] usually requires bioavailability/bioequivalence (BA/BE) studies. These pharmacokinetic clinical studies are designed to demonstrate comparable bioavailability or bioequivalence, defined as the absence of a significant difference in the rate and extent to which the active substance in pharmaceutical equivalent or pharmaceutical alternative medicinal products becomes available at the site of drug action when administered at the same molar dose under similar conditions [1]. Claiming bioequivalence between two products assumes an equivalent therapeutic efficacy and safety.

When companies are uncertain whether the potential of a new formulation is bioequivalent to a so called Reference product, it is usual to carry out downsized pilot studies as a gatekeeping in vivo strategy to decide whether or not to move forward with a full-size pivotal study [3,4,5].

Pilot studies data are usually analyzed similarly to pivotal studies, using the average bioequivalence approach, given that no formal methodologies are provided in the guidelines. However, due to the low number of subjects usually enrolled, the results obtained from these studies are difficult to interpret, particularly when the inter-occasion (IOV) or intra-subject variability is high, as the point estimate obtained for the means ratio may not be close to the real population value [5,6]. Consequently, pilot studies are considered underpowered studies.

In a previous work, authors have suggested a decision tree to be applied for the analysis of data from pilot BA/BE studies, which included the use of an alternative approach to the average bioequivalence, i.e., the similarity factor ƒ_2_ applied to the comparison of the geometric means (G_mean_) of plasma concentration–time profiles [3]. A cut off of 35 for the G_mean_ ƒ_2_ factor has been proposed to conclude on a potential similarity between the Test and Reference formulations on the absorption rate (as assessed by the maximum observed concentration [C_max_]), which is regulatorily required to be demonstrated in pivotal BA/BE studies [3]. For the tested simulated scenarios, this cut off demonstrated a good relationship between avoiding type I error (which represents the probability of erroneously conclude bioequivalence, known as consumer’s risk) and type II error (which represents the probability of erroneously conclude bioinequivalence, known as producer’s risk). However, the method was tested in ideal simulated scenarios, i.e., assuming either completely equal Test and Reference formulations (truly bioequivalent with a true Test-to-Reference Geometric Least Square Means [LSM] ratio [GMR] of 100%), or completely different formulations (truly bioinequivalent with a true GMR of 70%) [3].

Considering that during drug product development, less favorable GMRs are commonly expected, in this work, the authors aimed to further investigate the proposed G_mean_ ƒ_2_ factor in comparison with the standard average bioequivalence in more extreme and realistic scenarios, in order to better derive conclusions on the potential of this analysis method to be applied into pilot BA/BE studies. Hence, two major scenarios are tested:The Test product presents a lower bioavailability (BA) than the Reference product, with a true GMR of 90% (truly bioequivalent formulations) and 80% (truly bioinequivalent formulations).The Test product presents a higher bioavailability than the Reference product, with a true GMR of 111% (truly bioequivalent formulations) and 125% (truly bioinequivalent formulations).

For each of the two major scenarios tested, several pilot BA/BE crossover studies were simulated through population pharmacokinetic modelling, accounting different IOV levels. Method’ performance was measured with a confusion matrix.

## 2. Materials and Methods

For each major scenario, a total of 140,000 BA/BE crossover trials were simulated, corresponding to 5,880,000 different simulated concentration–time profiles per major scenario. For each major scenario, simulations were performed using two different Test-to-Reference ratios of the mean population values for the absorption rate constant (k_a_), different sample sizes and different IOV levels for the volume of distribution (V). In all simulations, a fixed value was used for inter-individual variability (IIV) (Figure 1).

Trial simulations and statistical analysis were performed with R version 4.0.3 (R Foundation for Scientific Computing, Vienna, Austria, 2013).

### 2.1. Study Design and Pharmacokinetic Simulation

Two-treatment (Test and Reference), two-sequence (Sequence 1 and Sequence 2), two-period crossover (2 × 2 × 2) studies were simulated (Figure 1—Study Design) as described by Henriques et al. (2023) [3]. A range of 12–30 simulated subjects per study were randomized and administered a single 50 mg oral dose of either Test or Reference products, separated by a washout of 7 days (Figure 1—Study Design).

As in the previous work [3], oral drug absorption and disposition were described using a one-compartmental model with first-order absorption and first-order elimination, defined through ordinary differential equations (ODE), parameterized with micro constants (Figure 1—Structural Model, and Equations (1) and (2), and considering a log-normal additive experimental error of 10% (Equation (3)) [3,7].
(1)$$\left\{\;\begin{array}{c} \frac{{dA}_{GI}}{dt}=-k_{a}∙A_{GI}\\ \frac{{dA}_{1}}{dt}=k_{a}∙A_{GI}-k_{e}∙A_{1} \end{array}\right.$$
(2)$$C=\frac{A_{1}}{V}$$
(3)$$Y=f\left(\theta ;x\right)\cdot e^{\varepsilon }\;⟺\;\log Y=logf\left(\theta ;x\right)+\varepsilon$$

A population pharmacokinetic modelling and simulation approach was used, computed by ‘Simulx’, a function of the ‘mlxR’ package version 4.1.3 (Monolix version 2019R2, Lixoft, Antony, France).

Each simulated pharmacokinetic profile comprised 20 simulated plasma samples, at time of dose (time 0) and at 0.25, 0.50, 0.75, 1.00, 1.50, 1.75, 2.00, 2.25, 2.50, 2.75, 3.00, 3.25, 3.50, 3.75, 4.00, 6.00, 8.00, 12.00, and 24.00 h after dose.

Regarding the compartmental model parameter k_a_, and for the scenario where the Test product showed a higher bioavailability than the Reference product, a fixed mean population value of 1.22 h^−1^ was assumed for the Reference product, and 0.732 h^−1^ (truly bioequivalent formulations) or 0.484 h^−1^ (truly bioinequivalent formulations) was assumed for the Test product. These k_a_ values were expected to provide a true GMR of approximately 90% and 80% for truly bioequivalent and truly bioinequivalent formulations, respectively (Figure 1—Covariate Model).

For the scenario where the Test product shows a lower bioavailability than the Reference product, a fixed mean population value of 1.22 h^−1^ was assumed for the Test product, and 0.732 h^−1^ (truly bioequivalent formulations) or 0.484 h^−1^ (truly bioinequivalent formulations) was assumed for the Reference product. These k_a_ values were expected to provide a true GMR of approximately 111% and 125% for truly bioequivalent and truly bioinequivalent formulations, respectively (Figure 1—Covariate Model).

For k_e_ and F model parameters, a mean population value of 0.150 h^−1^ for k_e_ and of 0.9 for F was assumed in all simulation scenarios.

Previous results showed that IOV in V was the variability identified with highest impact on the evaluation of C_max_ bioequivalence metric. The variability tested for the other model parameters had no relevant impact [3]. Therefore, in the present study, only IOV for V was included in the model. 

For each individual and occasion, V was generated considering a mean population value of 58.8 L, a log-normal distribution, a 30% IIV, and one of the following seven (7) different levels of IOV: (i) 0%, (ii) 10%, (iii) 20%, (iv) 30%, (v) 40%, (vi) 50%, and (vii) 60% (Figure 1—Statistical Model, Equation (4)). The impact of IIV was not assessed, as this variability was not expected to provide differences in the statistical analysis results, since it was suppressed by using a crossover design, as shown in previous simulations [3].
(4)$$\log\left(\Psi_{i}\right)\sim N\left(\log\left({\bar{\Psi }}_{i}\right),\;\left(\omega^{2},\;\gamma^{2}\right)\right)\;⟺\;\Psi_{i}={\bar{\Psi }}_{i}\cdot e^{\eta_{i}+\kappa_{i}},\;\mathrm{where}\;\eta_{i}\;\sim N\left(0,\omega^{2}\right)\;\mathrm{and}\;\kappa_{i}\;\sim N\left(0,\gamma^{2}\right)$$

Within each group of simulations and for each variability scenario, 1000 bioequivalence crossover trials were simulated. As in previous work, simulations only studied the effect of variability on the bioequivalence of C_max_ [3].

### 2.2. Simulation Bioequivalence Analysis

Simulation bioequivalence analysis and measure of methods’ performance were performed as described by Henriques et al. (2023) [3].

For each simulated bioequivalence trial, C_max_ was calculated and analyzed using the average bioequivalence approach, i.e., simulations were considered bioequivalent when the Test-to-Reference GMR and corresponding 90% CI were within [80.00–125.00]% [1,2,8,9,10,11,12,13]. From the average bioequivalence, the intra-subject coefficient of variation (ISCV%) was estimated from the obtained mean square error [1,3,9,12]. The centrality of the C_max_ Test-to-Reference GMR within [90.00–111.11]% [3] was also tested.

As an alternative to the average bioequivalence approach, the arithmetic (A_mean_) and geometric (G_mean_) mean ƒ_2_ factor approaches were tested with a cut off of 35, as proposed by Henriques et al. (2023) [3]. By placing a cut off of 35 for the ƒ_2_ factor, a maximum difference of 20% between the concentration–time profiles until the Reference t_max_ was tested [3,14].

Likewise, for each variability and number of subjects’ simulation scenario, the performance of each bioequivalence evaluation method (average bioequivalence, centrality of the Test-to-Reference GMR, and A_mean_ and G_mean_ ƒ_2_ factors) was measured with a confusion matrix in terms of sensitivity/power (capacity of avoiding type II errors), specificity (capacity of avoiding type I errors), precision (identified bioequivalent simulations that are truly bioequivalent), negative predictive value (NPV, identified bioinequivalent simulations that are truly bioinequivalent), accuracy (true bioequivalent and bioinequivalent predictions), F_1_ (harmonic mean of sensitivity and precision), Matthews’ Correlation Coefficient (MCC, correlation between the truth and the method prediction), and Cohen’s Kappa (κ, agreement relative to what would be expected by chance) [3,15,16]. For each tested method, the confusion matrix performance results were graphically presented over the number of subjects, for each tested variability scenario. Sensitivity and specificity were also plotted over the tested IOV.

## 3. Results

### 3.1. Simulated Pharmacokinetic Data

The summary statistics of the simulated pharmacokinetic parameter V are presented in the Appendix A section, along with 90% CI of the simulated concentration–time profiles, and the summary statistics of the estimated pharmacokinetic metrics C_max_, t_max_ and AUC.

For the simulation scenarios where a lower k_a_ value for the Test product was assumed in comparison to the Reference product, the Test product showed, in the case of truly bioequivalent formulations, a G_mean_ value for C_max_ between 577 and 583 µg/L (Appendix A), which was reached between 1.0 and 6 h (median t_max_ = 2.75 h) (Appendix A), and a G_mean_ value for AUC_0–t_ between 4900 and 4930 µg·h/L (Appendix A). In the case of truly bioinequivalent formulations, the Test product showed a G_mean_ value for C_max_ between 512 and 515 µg/L (Appendix A), which was reached between 1.5 and 8 h (median t_max_ = 3.25 h) (Appendix A), and a G_mean_ value for AUC_0–t_ between 4880 and 4900 µg·h/L (Appendix A). For both truly bioequivalent and truly bioinequivalent formulations, the Reference product demonstrated a G_mean_ value for C_max_ between 641 and 649 µg/L (Appendix A), which was reached between 0.75 and 4 h (median t_max_ = 2.25 h) (Appendix A), and a G_mean_ value for AUC_0–t_ between 4938 and 5000 µg·h/L (Appendix A).

For the simulation scenarios where a higher k_a_ value for the Test product was assumed in comparison to the Reference product, for both truly bioequivalent and truly bioinequivalent formulations, the Test product demonstrated a G_mean_ value for C_max_ between 639 and 646 µg/L (Appendix A), which was reached between 0.75 and 4 h (median t_max_ = 2.25 h) (Appendix A). The Reference product demonstrated, in the case of truly bioequivalent formulations, a G_mean_ value for C_max_ between 577 and 583 µg/L (Appendix A), which was reached between 1.0 and 6 h (median t_max_ = 2.75 h) (Appendix A), and a G_mean_ value for AUC_0–t_ between 4900 and 4950 µg·h/L (Appendix A). In the case of truly bioinequivalent formulations, the Reference product demonstrated a G_mean_ value for C_max_ between 511 and 515 µg/L (Appendix A), which was reached between 1.5 and 8 h (median t_max_ = 3.25 h) (Appendix A), and a G_mean_ value for AUC_0-t_ between 4867 and 4904 µg·h/L (Appendix A).

For V, C_max_, and AUC, the 95% CIs for G_mean_ are tightened, assuring an appropriate number of simulations per scenario. Moreover, the estimated geometric coefficient of variation (GCV%) results from the IIV and IOV components.

No differences were found for the apparent elimination half-life of the different simulated formulations, t_½_ ≈ 4.6 h.

### 3.2. Bioequivalence Evaluation

As planned, for the scenario where the Test product presents a lower bioavailability than the Reference product, the simulations for truly bioequivalent and truly bioinequivalent formulations demonstrated a mean GMR of approximately 90% and 80%, respectively, while for the scenario where the Test product presents a higher bioavailability than the Reference product, the simulations for truly bioequivalent and truly bioinequivalent formulations demonstrated a mean GMR of approximately 111% and 125%, respectively, with a coefficient of variation (CV%) of approximately 2%, 4%, 7%, 10%, 13%, 17%, and 20% for the simulations with an IOV of 0%, 10%, 20%, 30%, 40%, 50%, and 60%, respectively (Figure 2).

For both major scenarios tested, a mean ISCV% of approximately 6%, 12%, 20%, 30%, 42%, 53%, and 65% was observed for simulations with an IOV of 0%, 10%, 20%, 30%, 40%, 50%, and 60%, respectively (Figure 3).

For the two major scenarios tested, a mean ƒ_2_ factor of 37 was observed for truly bioequivalent formulations and a mean ƒ_2_ factor of 24 was observed for truly bioinequivalent formulations, with a CV% of approximately 5%, 10%, 18%, 25%, 30%, 35%, and 40% for simulations with an IOV of 0%, 10%, 20%, 30%, 40%, 50%, and 60%, respectively. These values corroborate the use of a cut off of 35 for the ƒ_2_ metric to evaluate a potential bioequivalence between two formulations in terms of C_max_ (Figure 4).

Such as with previous simulations [3], an inverted V-shaped correlation between the ƒ_2_ factor and GMR was found (Figure 5). However, unlike previous simulations where the plot was centered on a GMR of 100% [3], in the current simulations the V shape was moved to the opposite direction of the true GMR. Such behavior is a consequence of the fact that the ƒ_2_ factor was based on the normalization of the mean concentrations of the Test and Reference until the Reference t_max_. For simulations where the Test product shows a lower bioavailability than the Reference product (true GMR of 80% and 90%), the Reference product presented a faster absorption, resulting in a Reference t_max_ < Test t_max_, and hence in a cut off of the normalization of the mean concentration curves earlier than the occurrence of the Test C_max_. Consequently, the number of timepoints used for the calculation of the ƒ_2_ factor was reduced, increasing the ƒ_2_ value. On the other hand, for simulations where the Test product showed a higher bioavailability than the Reference product (true GMR of 111% and 125%), the Reference product had a slower absorption, resulting in a Reference t_max_ > Test t_max_, and hence in a cut off of the normalization of the mean concentration curves after the C_max_ of the Test product was reached. Consequently, the number of timepoints used for the calculation of the ƒ_2_ factor was increased, decreasing the ƒ_2_ value. Such behavior did not affect the performance of the method.

For both major scenarios tested, and for the lowest tested variability (an IOV from 0% to 10%), average bioequivalence was shown to be the most sensitive method, being able to detect nearly 100% of the truly bioequivalent formulations simulated with a 0% IOV, and being able to detect approximately 78% to 99% of the truly bioequivalent formulations simulated with a 10% IOV, in studies with 12 or 30 subjects. On the other hand, A_mean_ and G_mean_ ƒ_2_ factor approaches were less sensitive, detecting approximately 88% to 93% of the truly bioequivalent formulations simulated with a 0% IOV, and approximately 70% to 80% of the truly bioequivalent formulations simulated with a 10% IOV, in studies with 12 or 30 subjects (Figure 6 and Table 1 for the Test product with a lower bioavailability than the Reference product, and Figure 7 and Table 2 for a Test product with a higher bioavailability than the Reference product).

However, for simulations with a higher IOV (IOV ≥ 20%), the A_mean_ and G_mean_ ƒ_2_ factor demonstrated a higher sensitivity than the standard average bioequivalence analysis.

The ability of the average bioequivalence method to detect truly bioequivalent formulations decreased greatly, towards approximately 35% to 70% with 20% IOV, 10% to 40% with 30% IOV, and 2% to 20% with 40% IOV, in studies with 12 or 30 subjects, respectively. For an IOV greater than 50%, the sensitivity of the method was inferior to an IOV of 10% (Figure 6 and Table 1 for a Test product with a lower bioavailability than the Reference product, and Figure 7 and Table 2 for a Test product with a higher bioavailability than the Reference product).

The sensitivity/power of the A_mean_ and G_mean_ ƒ_2_ factor decreased as well with the increment of IOV, but not so steeply as with the standard method, allowing this alternative approach to demonstrate a superior sensitivity in the tested scenarios. For simulations with 20% IOV, the ƒ_2_ factor correctly identified around 60% to 70% of the truly bioequivalent formulations, while for simulations within 30% to 60% IOV, the A_mean_ and G_mean_ ƒ_2_ factor allowed nearly 50% to 60% of the truly bioequivalent formulations to be correctly identified (Figure 6 and Table 1 for a Test product with a lower bioavailability than the Reference product, and Figure 7 and Table 2 for a Test product with a higher bioavailability than the Reference product).

Regarding the specificity, as expected the average bioequivalence was suitable for avoiding the identification of false bioequivalent formulations and maintaining the type I error around 5%, irrespective of the sample size and IOV. The A_mean_ and G_mean_ ƒ_2_ factor approaches performed well in avoiding type I errors for an IOV < 40%. However, these approaches inflated type I errors when the IOV increased above 40%. For an IOV of 40%, a type I error < 5% was reached for trials simulated with 14 subjects. For an IOV of 50%, a type I error < 10% was reached for trials simulated with 16 subjects, decreasing to <5% for trials simulated with 24 subjects. For an IOV of 60%, a type I error < 10% was reached for trials simulated with 20 subjects, and the type I error was close to 5% for trials simulated with 28 subjects (Figure 8 and Table 1 for a Test product with a lower bioavailability than the Reference product, and Figure 9 and Table 2 for a Test product with a higher bioavailability than the Reference product).

For each of the two major scenarios, precision, NPV, accuracy, MCC, F_1_ and κ were also calculated in order to better understand the potentiality of each evaluation method in pilot BA/BE trials (Figure 10 and Table 1 for a Test product with a lower bioavailability than the Reference product, and Figure 11 and Table 2 for a Test product with a higher bioavailability than the Reference product).

The ƒ_2_ factor was always the most precise method, i.e., the method for which the identified bioequivalent formulations were more probable to be truly bioequivalent (Figure 10 and Table 1 for a Test product with a lower bioavailability than the Reference product, and Figure 11 and Table 2 for a Test product with a higher bioavailability than the Reference product).

Moreover, for higher-variability scenarios (an IOV ≥ 20%), the ƒ_2_ method was also the most reliable method for the identification of truly bioinequivalent formulations, i.e., the methodology with a higher NPV. For this method, the NPV varied little with the increment of subjects within the same variability scenarios (approximately 90% for 0% IOV, 80% for 10% IOV, and 70% for an IOV ≥ 20%). However, for lower variabilities (an IOV < 20%), the average bioequivalence was the method with a higher NPV (Figure 10 and Table 1 for a Test product with a lower bioavailability than the Reference product, and Figure 11 and Table 2 for a Test product with a higher bioavailability than the Reference product).

For an IOV ≥ 20%, the ƒ_2_ factor was also the most accurate methodology, showing a similar accuracy, despite the increase in sample size, within each simulated variability scenario (approximately 100% for 0% IOV, 90% for 10% IOV, 80% for an IOV within 20% and 30%, and 70% for an IOV within 40% and 60%) (Figure 10 and Table 1 for a Test product with a lower bioavailability than the Reference product, and Figure 11 and Table 2 for a Test product with a higher bioavailability than the Reference product).

Pondering simultaneously sensitivity and precision, the average bioequivalence was the method with lowest F_1_. On the other hand, the ƒ_2_ factor method could maintain a harmonic mean between sensitivity and precision, with an F_1_ of approximately 100% for a 0% IOV, 80% for an IOV within 10% and 20%, 70% for an IOV within 30% and 40%, and 60% for an IOV within 50% and 60% (Figure 10 and Table 1 for a Test product with a lower bioavailability than the Reference product, and Figure 11 and Table 2 for a Test product with a higher bioavailability than the Reference product).

Considering the correlation between the true classes and the predicted labels, once again, the average bioequivalence was the method that scored lower, and the ƒ_2_ factor was again the most superior method, with an MCC of approximately 90% for 0% IOV, 80% for an IOV within 10%, 70% for a 20% IOV, 60% for an IOV within 30% and 40%, and 50% for an IOV within 50% and 60% (Figure 10 and Table 1 for a Test product with a lower bioavailability than the Reference product, and Figure 11 and Table 2 for a Test product with a higher bioavailability than the Reference product).

Additionally, the study of the distribution of the calculated ƒ_2_ values for truly bioequivalent and truly bioinequivalent studies could also improve the certainty of the obtained results. These simulations showed that nearly 100% of the ƒ_2_ values above or equal to 50 (corresponding to a 10% difference between Test and Reference products [3,14]) were true positives (i.e., precision), irrespective of the IOV. Moreover, until a 40% IOV, more than 90% of the ƒ_2_ values above or equal to 41 (corresponding to a 15% difference between the Test and Reference products [3,14]) were true positives, and for an IOV within 50% to 60%, the precision of an ƒ_2_ above or equal to 41 was above 80%. For ƒ_2_ factors above or equal to 35, the precision was above 90% for simulations below a 20% IOV, and was above 80% for an IOV of 30%. For an IOV above 40%, more than 60% of the ƒ_2_ values above or equal to 35 were true positives (Figure 12). The combination of the ƒ_2_ factor method with the centrality of the GMR did not improve the precision of the method.

Simulations also elucidated that, for truly bioequivalent formulations where Test and Reference differ by a maximum of 10% on C_max_, the probability of the point estimate (GMR) being centered within [90.00–111.11]% is around 60% in baseline studies without a tested IOV. This probability linearly decreases towards 32.2% to 42.2% for simulations with a 60% IOV, in trials with 12 or 30 subjects. For truly bioinequivalent formulations where Test and Reference differ by at least 20% on C_max_, the probability of a false centered GMR can be around 20% for the higher tested IOV. Nevertheless, the probability of a centered GMR to indicate a truly bioequivalent simulation (precision) was nearly 100% for simulations with a 10% IOV, within approximately 80% to 90% for simulations within 20% and 30% IOV, and approximately within 60% to 70% for simulations within 40% and 60% IOV (Figure 10 and Table 1 for a Test product with a lower bioavailability than the Reference product, and Figure 11 and Table 2 for a Test product with a higher bioavailability than the Reference product).

## 4. Discussion

The average bioequivalence method proved to be the most sensitive method in the simulations performed for the lowest tested variability scenarios (an IOV from 0% to 10%), in both major scenarios tested. Based on Figure 6 and Table 1 results (regarding a Test product with a lower bioavailability than the Reference product), considering an IOV of 0%, the sensitivity was ≥99.4% for the average bioequivalence method, while for the other methods it was ≤94.4%. Considering an IOV of 10%, the sensitivity was ≥78.4% for the average bioequivalence method, while for the other methods it was ≤79.5%. Based on Figure 7 and Table 2 results (regarding a Test product with a higher bioavailability than the Reference product), for an IOV of 0% the sensitivity was 100% for the average bioequivalence method, and it was ≤97.0% for the other methods. Considering an IOV of 10%, the sensitivity was ≥77.7% for the average bioequivalence method and ≤77.6% for the other methods.

However, for an IOV ≥ 20%, A_mean_ and G_mean_ ƒ_2_ factor approaches have shown a higher sensitivity/power than the standard average bioequivalence analysis (Figure 6 and Table 1 for a Test product with a lower bioavailability than the Reference product, and Figure 7 and Table 2 for a Test product with a higher bioavailability than the Reference product). Based on Table 1 results and considering an IOV of 20%, the sensitivity values derived for A_mean_ and G_mean_ ƒ_2_ factor methods were more concise (between 60.2 and 68.8%), while for the average bioequivalence method, derived sensitivity ranged from 35.2 to 74.3%. For an IOV ≥ 30%, the derived sensitivity for A_mean_ and G_mean_ ƒ_2_ factor methods was shown to be ≥49.7%, as it was always considerably higher than the sensitivity for the average bioequivalence method for each scenario. Based on Table 2 results and considering an IOV of 20%, the sensitivity values derived for A_mean_ and G_mean_ ƒ_2_ factor methods were also more concise (between 62.5 and 63.5%), while for the average bioequivalence method, derived sensitivity ranged from 36.2 to 67.6%. For an IOV ≥ 30%, derived sensitivity for A_mean_ and G_mean_ ƒ_2_ factor methods was shown to be ≥46.9%, and was also always considerably higher than the sensitivity for the average bioequivalence method for each scenario.

The sensitivity/power of the average bioequivalence method demonstrated a sigmoidal decrease, from ≈100% to ≈0%, in function of IOV, with slopes decreasing considerably with the increment in the number of subjects per trial (Figure 6 and Figure 7). Thus, such results confirm the high sensitivity of the method to the increment on the number of subjects.

On the other hand, and based on the same figures, the sensitivity/power of the A_mean_ and G_mean_ ƒ_2_ factor method decreased exponentially from ≈90% to a plateau of ≈60%, showing no meaningful differences in the slope with the increase in the number of subjects per trial. Moreover, for higher variabilities (an IOV > 30%), the increase in IOV did not correspond to a higher decrease in sensitivity. Considering that this method relies on the mean profile of the concentration–time curves, the increase in sample size does not greatly increase the sensitivity of the ƒ_2_ factor method. Nevertheless, for a higher variability (an IOV > 40%), the increase in the sample size can reduce the rate of type I errors, as assessed by the specificity. For an IOV of 40%, a type I error < 5% was reached for trials simulated with 14 subjects, and for an IOV of 50%, type I error < 5% was reached for trials simulated with 24 subjects. For an IOV of 60%, type I error was close to 5% for trials simulated with 28 subjects (Figure 8 for a Test product with a lower bioavailability than the Reference product, and Figure 9 for a Test product with a higher bioavailability than the Reference product).

Based on Figure 10 and Table 1 for a Test product with a lower bioavailability than the Reference product, as well as on Figure 11 and Table 2 for a Test product with a higher bioavailability than the Reference product, the ƒ_2_ factor was always the most precise method and the method that demonstrated the best relationship between sensitivity/power and precision (F_1_). The ƒ_2_ factor was also the method with the best correlation between reality and the method prediction, as defined by accuracy, MCC, and κ. Additionally, the ƒ_2_ factor was the most reliable method for the identification of truly bioinequivalent formulations (NPV) when the variability was high (≤20%). Nevertheless, for lower variability scenarios (<20%), bioinequivalent results were more reliable for the average bioequivalence method.

Pondering the observations from the current simulations, the authors refined the previously purposed decision tree for the analysis of data from pilot BA/BE studies [3] (Figure 13). This decision tree is thought to be able to assist companies on their decision to move forward with a full-size pivotal study, for drugs following a one compartment model, with median t_max_ ranging from 0.75 to 8 h, a mean elimination half-life of approximately 4.6 h, and a mean volume of distribution of approximately 60 L, as the limits of tested scenarios. As before, for drug products with a known ISCV% below 20%, the authors propose the estimation of the sample size for a pilot study assuming a GMR of 100%, a power of 80%, and an α of 0.05 [3].

However, for cases of higher ISCV% or unknow variability, the authors propose the use of a fixed sample size of 20 subjects in the current work, as the use of higher sample sizes was not shown to increase the study power meaningfully, but was sufficient to avoid substantial type I errors. Regarding the analysis of data from pilot studies, the authors keep the methodology previously proposed [3], i.e., to initially analyze the data using the average bioequivalence approach. For the case in which the calculated GMR and the corresponding 90% CI are not within [80.00–125.00]%, the alternative G_mean_ ƒ_2_ factor method should be used with a cut off of 35 (Figure 13), as it was shown to be a valuable indicator of the potentiality of the Test formulation to be bioequivalent in terms of C_max_ with a Reference product [3]. Nevertheless, in this work the authors redefine the interpretation of the G_mean_ ƒ_2_ factor results based on the greatness of the calculated value (Figure 12 and Figure 13):If the ƒ_2_ factor is above or equal to 35 (corresponding to a difference of 20% between Test and Reference concentration–time profiles until the Reference t_max_), the confidence to proceed to a pivotal study is higher than 90% when ISCV% is lower or equal to 20%; the confidence is higher than 80% when ISCV% is within 20% and 30%; and the confidence is higher than 60% when ISCV% is higher than 40%.If the ƒ_2_ factor is above or equal to 41 (corresponding to a difference of 15% between Test and Reference concentration–time profiles until the Reference t_max_), the confidence to proceed to a pivotal study is higher than 90% for ISCV% until 40%, and higher than 80% for ISCV% within 50% to 60%.If the ƒ_2_ factor is above or equal to 50 (corresponding to a difference of 10% between Test and Reference concentration–time profiles until the Reference t_max_), the probability of the Test product to be truly bioequivalent to the Reference product in terms of C_max_, i.e., the confidence to proceed to a pivotal study, is higher than 90%, irrespective of the ISCV%.

## 5. Conclusions

Due to the reduced sample size, and consequently being underpowered, the results derived from pilot BA/BE trials performed with drug/drug products showing a considerable variability (ISCV% > 20%) are dubious, and consequently the conclusions affecting the evaluation of the potential of a Test formulation to be bioequivalent to a Reference formulation are uncertain. Therefore, the authors have proposed the G_mean_ ƒ_2_ as an alternative approach to the average bioequivalence methodology that is generally applied to pilot studies to access the rate of drug absorption [3]. The G_mean_ ƒ_2_ was shown to be capable of overcoming and reducing the uncertainty of these underpowered studies, which can meaningfully aid pharmaceutical companies in the decision to go forward with pivotal bioequivalence studies [3].

In this project, the authors continued their previous work [3] and performed simulations in more extreme scenarios, using a true GMR of 90% or 111% for truly bioequivalent formulations, and 80% or 125% for truly bioinequivalent formulations, in order to better derive conclusions on the potential of this analysis method in more realistic and extreme scenarios.

A redefined decision tree is proposed, suggesting a fixed sample size of 20 subjects for pilot studies in the case of an ISCV% > 20% or which is unknown and the assessment of study results through the average bioequivalence analysis and additionally through the G_mean_ ƒ_2_ factor in the case of the 90% CI for GMR, which is outside the regulatory acceptance bioequivalence interval of [80.00–125.00]% (Figure 13). Using this alternative approach, the certainty levels to proceed for pivotal studies depending on G_mean_ ƒ_2_ values and variability scenarios tested (20–60% IOV) were assessed, which is expected to be helpful in terms of the decision to go forward with pivotal bioequivalence studies.

## Figures and Tables

**Figure 1 pharmaceutics-15-02498-f001:**
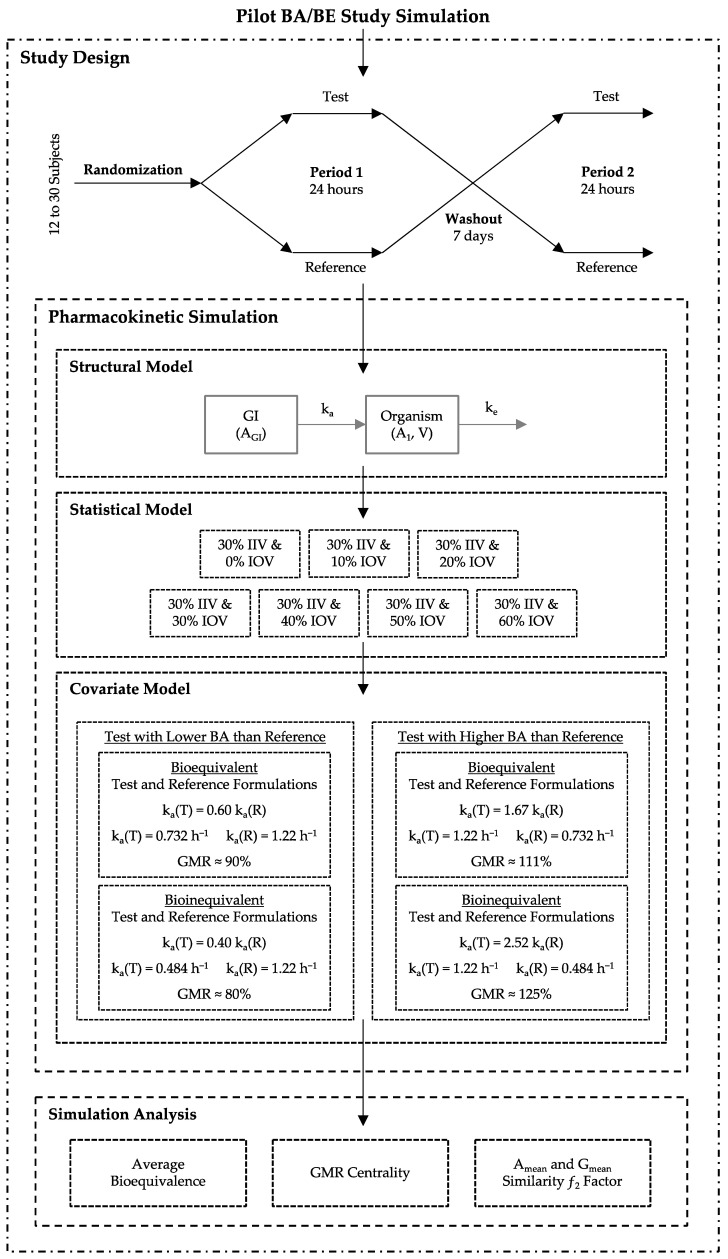
Simulation scheme for pilot bioavailability/bioequivalence (BA/BE) trials.

**Figure 2 pharmaceutics-15-02498-f002:**
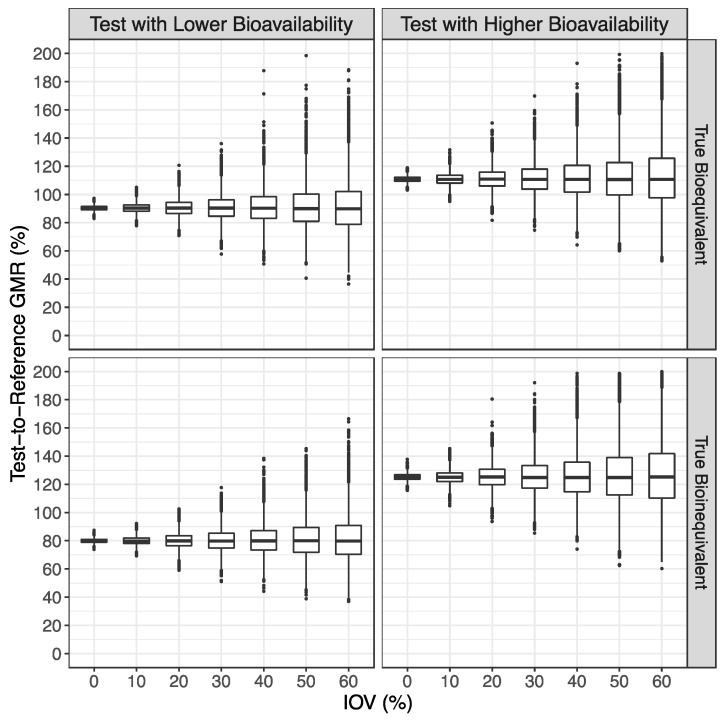
Distribution of the Test-to-Reference Geometric Least Square Means Ratio (GMR), estimated from the average bioequivalence method, in the form of box plots.

**Figure 3 pharmaceutics-15-02498-f003:**
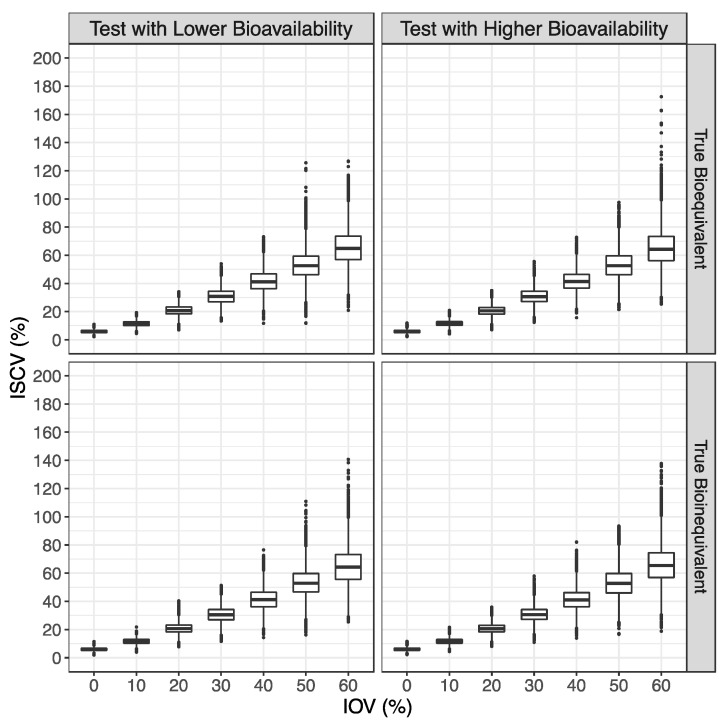
Distribution of the intra-subject coefficient of variation (ISCV%), estimated from the average bioequivalence method, in the form of box plots.

**Figure 4 pharmaceutics-15-02498-f004:**
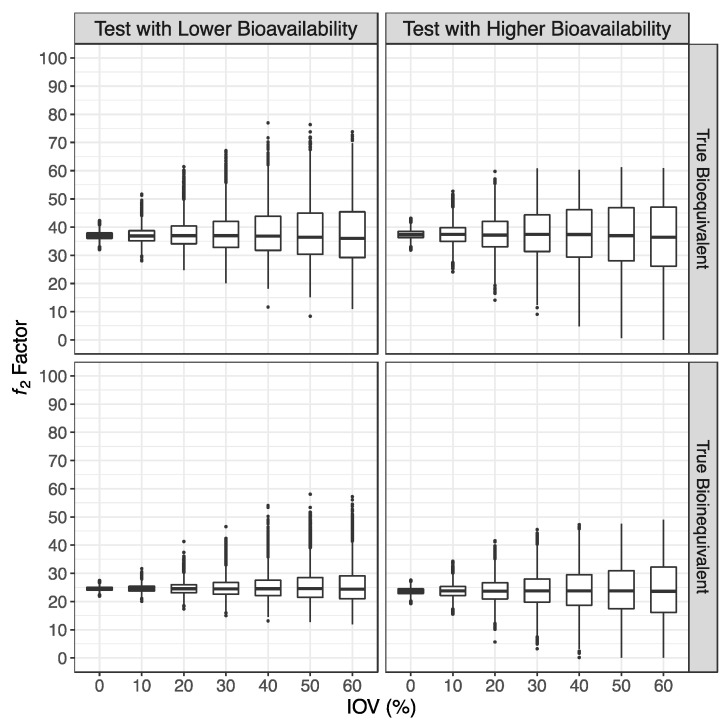
Distribution of the calculated ƒ_2_ factor, in the form of box plots.

**Figure 5 pharmaceutics-15-02498-f005:**
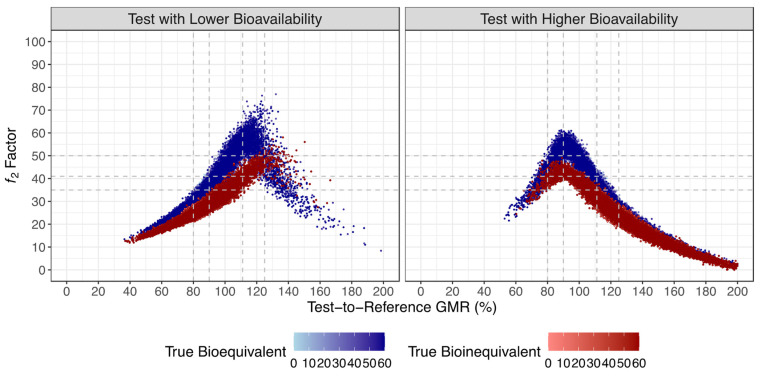
Relationship between G_mean_ *f*_2_ factor and Test-to-Reference GMR (above) for all simulated truly bioequivalent (blue) and truly bioinequivalent (red) studies. Vertical dotted lines correspond to 10% and 20% difference between Test and Reference formulations, tested by the average bioequivalence approach. Horizontal dotted lines correspond to ƒ_2_ values of 50, 41, and 35.

**Figure 6 pharmaceutics-15-02498-f006:**
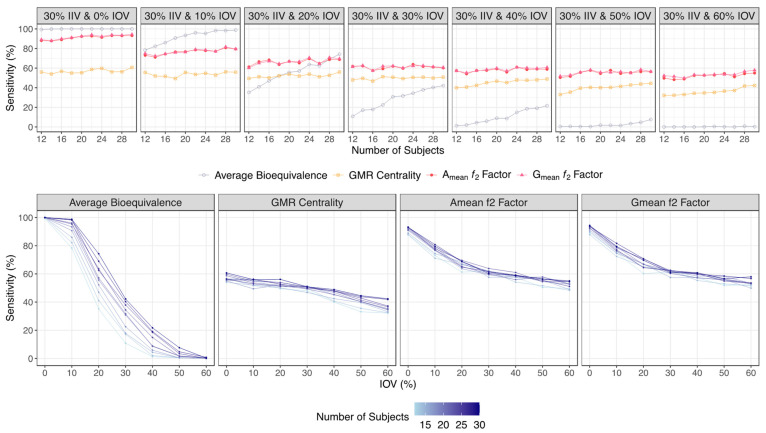
Variation in sensitivity/power for the bioequivalence evaluation methods (average bioequivalence, centrality of the Test-to-Reference GMR, and A_mean_ and G_mean_ ƒ_2_ factor evaluated with a cut off of 35) as function of the number of subjects for each tested variability (above) and as function of inter-occasion variability (below), considering a Test product with a lower bioavailability than the Reference product (i.e., true GMR of 90%).

**Figure 7 pharmaceutics-15-02498-f007:**
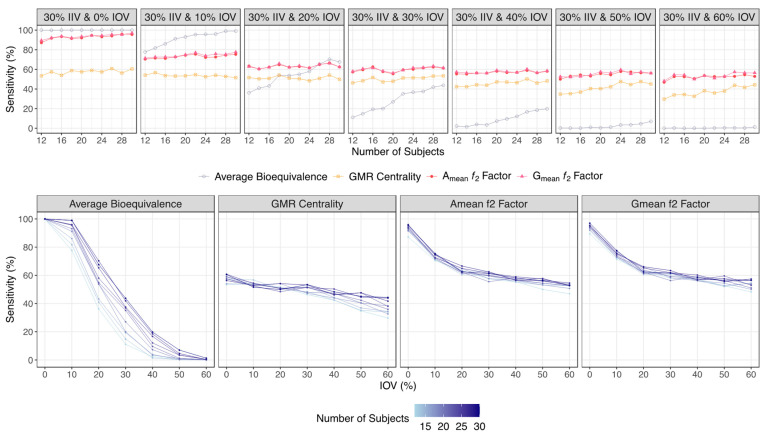
Variation in sensitivity/power for the bioequivalence evaluation methods (average bioequivalence, centrality of the Test-to-Reference GMR, and A_mean_ and G_mean_ ƒ_2_ factor evaluated with a cut off of 35) as function of the number of subjects for each tested variability (above) and as function of inter-occasion variability (below), considering a Test product with a higher bioavailability than the Reference product (i.e., true GMR of 111%).

**Figure 8 pharmaceutics-15-02498-f008:**
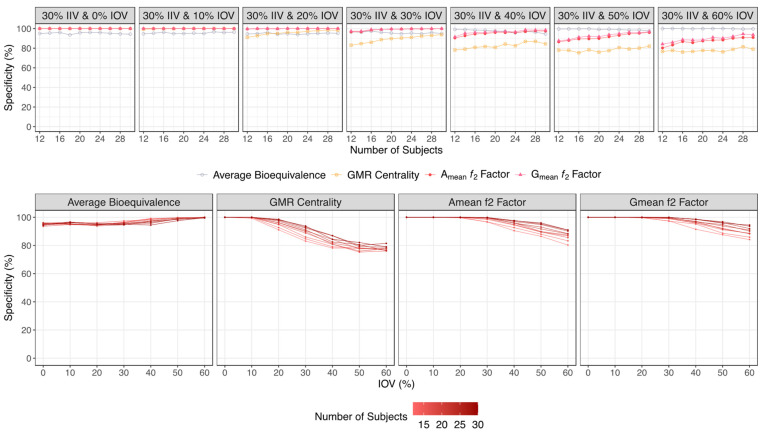
Variation in specificity for the bioequivalence evaluation methods (average bioequivalence, centrality of the Test-to-Reference GMR, and A_mean_ and G_mean_ ƒ_2_ factor evaluated with a cut off of 35) as function of the number of subjects for each tested variability (above) and as function of inter-occasion variability (below), considering a Test product with a lower bioavailability than the Reference product (i.e., true GMR of 80%).

**Figure 9 pharmaceutics-15-02498-f009:**
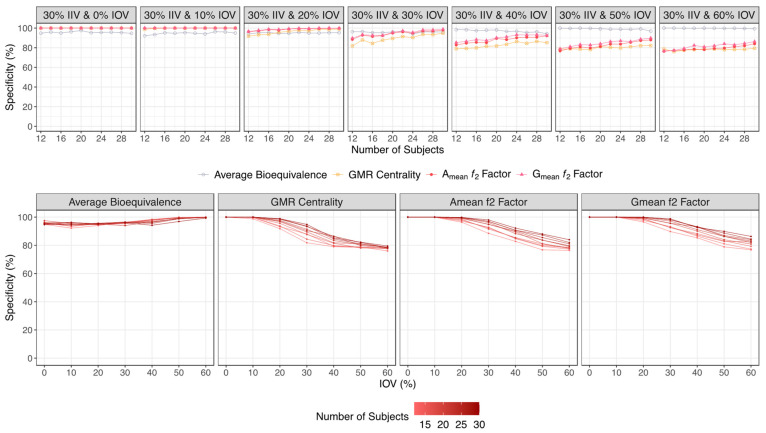
Variation in specificity for the bioequivalence evaluation methods (average bioequivalence, centrality of the Test-to-Reference GMR, and A_mean_ and G_mean_ ƒ_2_ factor evaluated with a cut off of 35) as function of the number of subjects for each tested variability (above) and as function of inter-occasion variability (below), considering a Test product with a higher bioavailability than the Reference product (i.e., true GMR of 125%).

**Figure 10 pharmaceutics-15-02498-f010:**
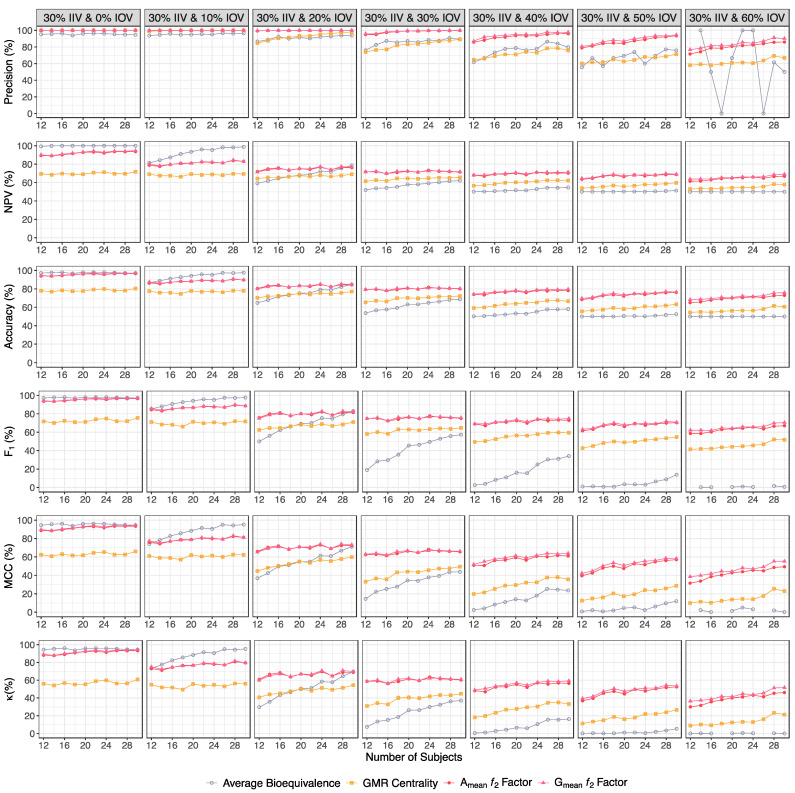
Variation in precision, negative predictive value (NPV), accuracy, F_1_, Matthews’ Correlation Coefficient (MCC), and Cohen’s Kappa (κ) for the bioequivalence evaluation methods (average bioequivalence, centrality of the Test-to-Reference GMR, and A_mean_ and G_mean_ ƒ_2_ factor evaluated with a cut off of 35) as function of the number of subjects for each tested variability, considering a Test product with a lower bioavailability than the Reference product (i.e., true GMR of 90% and 80%).

**Figure 11 pharmaceutics-15-02498-f011:**
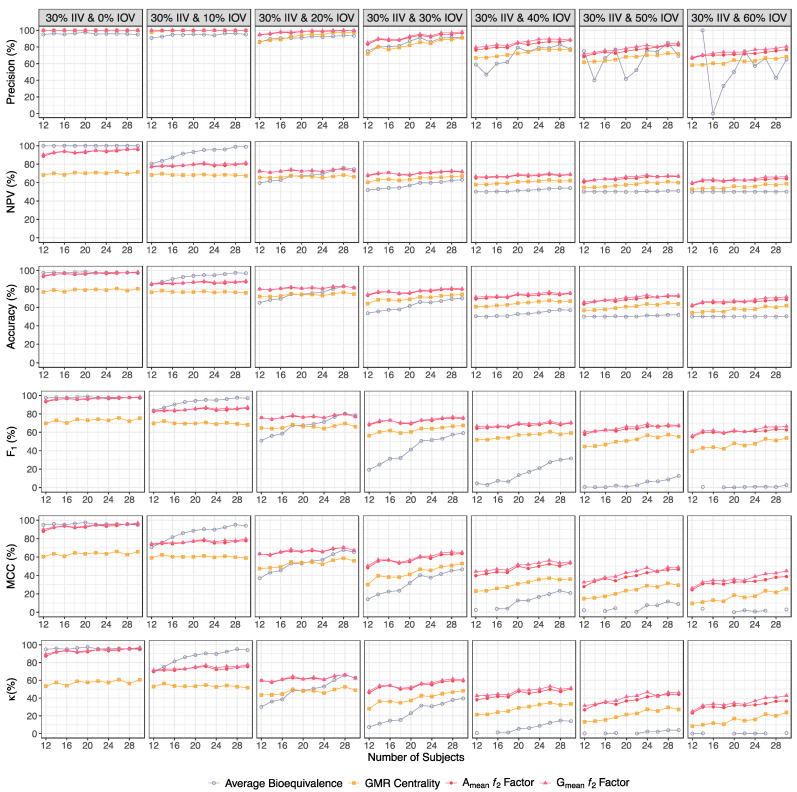
Variation in precision, negative predictive value (NPV), accuracy, F_1_, Matthews’ Correlation Coefficient (MCC), and Cohen’s Kappa (κ) for the bioequivalence evaluation methods (average bioequivalence, centrality of the Test-to-Reference GMR, and A_mean_ and G_mean_ ƒ_2_ factor evaluated with a cut off of 35) as function of the number of subjects for each tested variability, considering a Test product with a higher bioavailability than the Reference product (i.e., true GMR of 111% and 125%).

**Figure 12 pharmaceutics-15-02498-f012:**
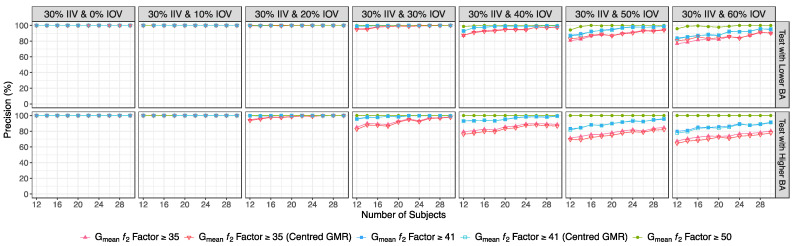
Variation in precision, for the G_mean_ ƒ_2_ factor evaluated with a cut off of 35, 41, and 50 as function of the number of subjects for each tested variability. An ƒ_2_ factor of 35, 41, and 50 corresponds to a difference of 20%, 15%, and 10%, respectively, between Test and Reference concentration time-profiles until the Reference t_max_.

**Figure 13 pharmaceutics-15-02498-f013:**
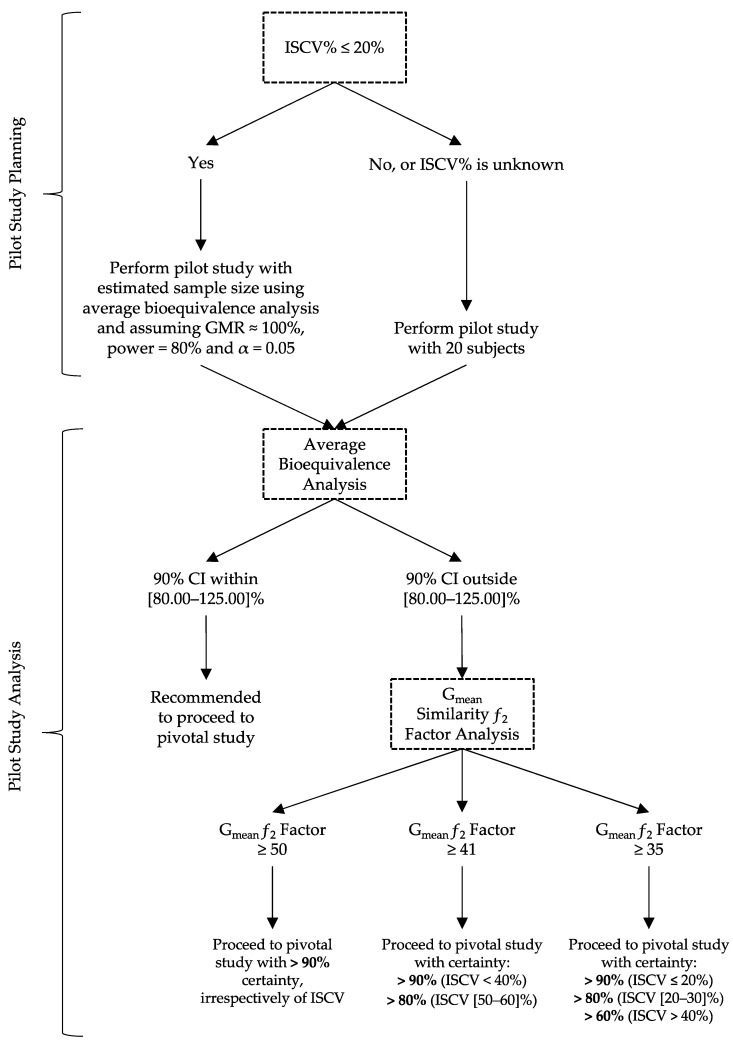
Newly proposed decision tree for planning and analysis of pilot BA/BE studies.

**Table 1 pharmaceutics-15-02498-t001:** Cross-tabulated matrix statistics calculated for each bioequivalence evaluation method (average bioequivalence, centrality of the Test-to-Reference GMR, and A_mean_ and G_mean_ ƒ_2_ factor evaluated with a cut off of 35) for each tested variability, considering a Test product with a lower bioavailability than the Reference product (i.e., true GMR of 90% and 80%).

	Average Bioequivalence	GMR Centrality	A_mean_ ƒ_2_ Factor	G_mean_ ƒ_2_ Factor
Sensitivity (%)				
30% IIV &0% IOV	99.4–100	56.0–60.7	88.1–93.2	89.1–94.4
30% IIV & 10% IOV	78.4–98.8	55.6–55.9	73.1–79.5	74.9–79.5
30% IIV &20% IOV	35.2–74.3	49.5–56.1	61.3–68.8	60.2–69.7
30% IIV &30% IOV	10.8–42.2	47.9–50.8	61.9–60.2	61.3–60.8
30% IIV &40% IOV	1.30–21.7	40.0–48.9	57.6–58.8	57.2–60.7
30% IIV &50% IOV	0.50–7.60	33.1–44.5	50.4–56.4	51.8–56.6
30% IIV &60% IOV	0.00–0.30	32.2–42.3	49.7–55.0	52.3–57.9
Type II Error (%)				
30% IIV &0% IOV	0.60–0.00	44.0–39.3	11.9–6.80	10.9–5.60
30% IIV &10% IOV	21.6–1.20	44.4–44.1	26.9–20.5	25.1–20.5
30% IIV &20% IOV	64.8–25.7	50.5–43.9	38.7–31.2	39.8–30.3
30% IIV &30% IOV	89.2–57.8	52.1–49.2	38.1–39.8	38.7–39.2
30% IIV &40% IOV	98.7–78.3	60.0–51.1	42.4–41.2	42.8–39.3
30% IIV &50% IOV	100–92.4	66.9–55.5	49.6–43.6	48.2–43.4
30% IIV &60% IOV	100–99.7	67.8–57.7	50.3–45.0	47.7–42.1
Specificity (%)				
30% IIV & 0% IOV	95.0–94.3	100	100	100
30% IIV & 10% IOV	94.7–96.4	99.3–100	100	100
30% IIV & 20% IOV	94.5–95.2	91.1–98.2	99.6–100	99.7–100
30% IIV & 30% IOV	96.6–95.0	83.1–93.8	96.7–100	97.4–100
30% IIV & 40% IOV	99.2–94.5	78.2–84.4	90.5–97.5	91.5–98.5
30% IIV & 50% IOV	100–97.6	78.1–82.1	86.5–96.0	87.6–96.8
30% IIV & 60% IOV	100–99.7	76.7–79.0	80.3–91.0	84.2–93.7
Type I Error (%)				
30% IIV & 0% IOV	5.00–5.70	0.00	0.00	0.00
30% IIV & 10% IOV	5.30–3.60	0.70–0.00	0.00	0.00
30% IIV & 20% IOV	5.50–4.80	8.90–1.80	0.40–0.00	0.30–0.00
30% IIV & 30% IOV	3.40–5.00	16.9–6.20	3.30–0.10	2.60–0.00
30% IIV & 40% IOV	0.80–5.50	21.8–15.6	9.5–2.50	8.50–1.50
30% IIV & 50% IOV	0.40–2.40	21.9–17.9	13.5–4.00	12.4–3.20
30% IIV & 60% IOV	0.00–0.30	23.3–21.0	19.7–9.00	15.8–5.50
Precision (%)				
30% IIV & 0% IOV	95.2–94.6	100	100	100
30% IIV & 10% IOV	93.7–96.5	98.8–100	100	100
30% IIV & 20% IOV	86.5–93.9	84.8–96.9	99.4–100	100–100
30% IIV & 30% IOV	76.1–89.4	73.9–89.1	94.9–100	95.9–100
30% IIV & 40% IOV	61.9–79.8	64.7–75.8	85.8–95.9	87.1–97.6
30% IIV & 50% IOV	55.6–76.0	60.2–71.3	78.9–93.4	80.7–94.6
30% IIV & 60% IOV	NC–50.0	58.0–66.8	71.6–85.9	76.8–90.2
NPV (%)				
30% IIV & 0% IOV	99.4–100	69.4–71.8	89.4–93.6	90.2–94.7
30% IIV & 10% IOV	81.4–98.8	69.1–69.4	78.8–83.0	79.9–83.0
30% IIV & 20% IOV	59.3–78.7	64.3–69.1	72.0–76.2	71.5–76.7
30% IIV & 30% IOV	52.0–62.2	61.5–65.6	71.7–71.5	71.6–71.8
30% IIV & 40% IOV	50.1–54.7	56.6–62.3	68.1–70.3	68.1–71.5
30% IIV & 50% IOV	50.0–51.4	53.9–59.7	63.6–68.8	64.5–69.0
30% IIV & 60% IOV	50.0	53.1–57.8	61.5–66.9	63.8–69.0
Accuracy (%)				
30% IIV & 0% IOV	97.2–97.2	78.0–80.4	94.1–96.6	94.6–97.2
30% IIV & 10% IOV	86.6–97.6	77.5–78.0	86.6–89.8	87.5–89.8
30% IIV & 20% IOV	64.9–84.8	70.3–77.2	80.5–84.4	80.0–84.9
30% IIV & 30% IOV	53.7–68.6	65.5–72.3	79.3–80.1	79.4–80.4
30% IIV & 40% IOV	50.3–58.1	59.1–66.7	74.1–78.2	74.4–79.6
30% IIV & 50% IOV	50.1–52.6	55.6–63.3	68.5–76.2	69.7–76.7
30% IIV & 60% IOV	50.0	54.5–60.7	65.0–73.0	68.3–75.8
F_1_ (%)				
30% IIV & 0% IOV	97.3–97.2	71.8–75.5	93.7–96.5	94.2–97.1
30% IIV & 10% IOV	85.4–97.6	71.1–71.7	84.5–88.6	85.6–88.6
30% IIV & 20% IOV	50.0–83.0	62.5–71.1	75.8–81.5	75.0–82.1
30% IIV & 30% IOV	18.9–57.3	58.1–64.7	74.9–75.1	74.8–75.6
30% IIV & 40% IOV	2.55–34.1	49.4–59.5	68.9–72.9	69.0–74.8
30% IIV & 50% IOV	0.99–13.8	42.7–54.8	61.5–70.3	63.1–70.8
30% IIV & 60% IOV	NC–0.60	41.4–51.8	58.7–67.1	62.2–70.5
MCC (%)				
30% IIV & 0% IOV	94.5–94.5	62.4–66.0	88.7–93.4	89.6–94.5
30% IIV & 10% IOV	74.1–95.2	61.0–62.3	75.9–81.2	77.4–81.2
30% IIV & 20% IOV	36.9–71.1	44.6–59.9	65.9–72.4	65.2–73.1
30% IIV & 30% IOV	14.4–43.8	33.1–49.4	62.5–65.5	62.9–66.1
30% IIV & 40% IOV	2.45–23.6	19.7–35.6	50.9–61.1	51.8–63.9
30% IIV & 50% IOV	0.75–11.9	12.5–28.7	39.6–57.1	42.2–58.3
30% IIV & 60% IOV	NC–0.00	9.9–22.9	31.5–49.3	38.5–55.3
κ (%)				
30% IIV & 0% IOV	94.4–94.3	56.0–60.7	88.1–93.2	89.1–94.4
30% IIV & 10% IOV	73.1–95.2	54.9–55.9	73.1–79.5	74.9–79.5
30% IIV & 20% IOV	29.7–69.5	40.6–54.3	60.9–68.8	59.9–69.7
30% IIV & 30% IOV	7.40–37.2	31.0–44.6	58.6–60.1	58.7–60.8
30% IIV & 40% IOV	0.50–16.2	18.2–33.3	48.1–56.3	48.7–59.2
30% IIV & 50% IOV	0.10–5.20	11.2–26.6	36.9–52.4	39.4–53.4
30% IIV & 60% IOV	0.00	8.90–21.3	30.0–46.0	36.5–51.6

Values represent the range calculated from simulated studies with 12 and 30 subjects. When statistics do not change between 12 and 30 subjects, unique values are presented instead of ranges. F_1_—harmonic mean of sensitivity and precision; κ—Cohen’s Kappa; MCC—Matthews’ correlation coefficient; NPV—negative predictive value. NC—not calculated.

**Table 2 pharmaceutics-15-02498-t002:** Cross-tabulated matrix statistics calculated for each bioequivalence evaluation method (average bioequivalence, centrality of the Test-to-Reference GMR, and A_mean_ and G_mean_ ƒ_2_ factor evaluated with a cut off of 35) for each tested variability, considering a Test product with a higher bioavailability than the Reference product (i.e., true GMR of 111% and 125%).

	Average Bioequivalence	GMR Centrality	A_mean_ ƒ_2_ Factor	G_mean_ ƒ_2_ Factor
Sensitivity (%)				
30% IIV & 0% IOV	100	53.4–60.5	87.2–95.6	89.5–97.0
30% IIV & 10% IOV	77.7–99.0	54.1–51.6	70.5–75.3	71.4–77.6
30% IIV & 20% IOV	36.2–67.6	51.7–49.9	63.5–62.5	62.8–62.5
30% IIV & 30% IOV	11.1–43.7	46.3–53.4	57.4–61.3	58.0–61.6
30% IIV & 40% IOV	2.30–19.8	42.4–48.3	55.4–58.1	57.3–58.6
30% IIV & 50% IOV	0.30–7.00	34.8–45.1	50.0–56.2	52.6–56.1
30% IIV & 60% IOV	0.00–1.30	29.7–44.3	46.9–52.9	48.2–56.5
Type II Error (%)				
30% IIV & 0% IOV	0.00	46.6–39.5	12.8–4.40	10.5–3.00
30% IIV & 10% IOV	22.3–1.00	45.9–48.4	29.5–24.7	28.6–22.4
30% IIV & 20% IOV	63.8–32.4	48.3–50.1	36.5–37.5	37.2–37.5
30% IIV & 30% IOV	88.9–56.3	53.7–46.6	42.6–38.7	42.0–38.4
30% IIV & 40% IOV	97.7–80.2	57.6–51.7	44.6–41.9	42.7–41.4
30% IIV & 50% IOV	100–93.0	65.2–54.9	50.0–43.8	47.4–43.9
30% IIV & 60% IOV	100–98.7	70.3–55.7	53.1–47.1	51.8–43.5
Specificity (%)				
30% IIV & 0% IOV	94.8–94.7	100	100	100
30% IIV & 10% IOV	92.2–95.0	98.8–100	100–100	100
30% IIV & 20% IOV	94.0–95.3	91.8–98.9	96.4–100	96.6–100
30% IIV & 30% IOV	96.3–96.0	81.8–94.8	88.5–98.0	89.8–98.8
30% IIV & 40% IOV	98.4–94.2	79.0–85.2	82.8–92.2	85.3–92.6
30% IIV & 50% IOV	100–96.9	78.5–82.2	76.8–88.0	78.9–89.9
30% IIV & 60% IOV	100–99.3	78.7–79.5	76.4–84.0	76.8–86.3
Type I Error (%)				
30% IIV & 0% IOV	5.20–5.30	0.00	0.00	0.00
30% IIV & 10% IOV	7.80–5.00	1.20–0.00	0.10–0.00	0.00
30% IIV & 20% IOV	6.00–4.70	8.20–1.10	3.60–0.20	3.40–0.10
30% IIV & 30% IOV	3.70–4.00	18.2–5.20	11.5–2.00	10.2–1.20
30% IIV & 40% IOV	1.60–5.80	21.0–14.8	17.2–7.80	14.7–7.40
30% IIV & 50% IOV	0.10–3.10	21.5–17.8	23.2–12.0	21.1–10.1
30% IIV & 60% IOV	0.00–0.70	21.3–20.5	23.6–16.0	23.2–13.7
Precision (%)				
30% IIV & 0% IOV	95.1–95.0	100	100	100
30% IIV & 10% IOV	90.9–95.2	97.8–100	100	100
30% IIV & 20% IOV	85.8–93.5	86.3–97.8	94.6–100	94.9–100
30% IIV & 30% IOV	75.0–91.6	71.8–91.1	83.3–96.8	85.0–98.1
30% IIV & 40% IOV	59.0–77.3	66.9–76.5	76.3–88.2	79.6–88.8
30% IIV & 50% IOV	75.0–69.3	61.8–71.7	68.3–82.4	71.4–84.7
30% IIV & 60% IOV	NC–65.0	58.2–68.4	66.5–76.8	67.5–80.5
NPV (%)				
30% IIV & 0% IOV	100	68.2–71.7	88.7–95.8	90.5–97.1
30% IIV & 10% IOV	80.5–99.0	68.3–67.4	77.2–80.2	77.8–81.7
30% IIV & 20% IOV	59.6–74.6	65.5–66.4	72.5–72.7	72.2–72.7
30% IIV & 30% IOV	52.0–63.0	60.4–67.0	67.5–71.7	68.1–72.0
30% IIV & 40% IOV	50.2–54.0	57.8–62.2	65.0–68.8	66.6–69.1
30% IIV & 50% IOV	50.1–51.0	54.6–60.0	60.6–66.8	62.5–67.2
30% IIV & 60% IOV	50.0–50.2	52.8–58.8	59.0–64.1	59.7–66.5
Accuracy (%)				
30% IIV & 0% IOV	97.4–97.4	76.7–80.3	93.6–97.8	94.8–98.5
30% IIV & 10% IOV	85.0–97.0	76.5–75.8	85.2–87.7	85.7–88.8
30% IIV & 20% IOV	65.1–81.5	71.8–74.4	80.0–81.2	79.7–81.2
30% IIV & 30% IOV	53.7–69.9	64.1–74.1	73.0–79.7	73.9–80.2
30% IIV & 40% IOV	50.4–57.0	60.7–66.8	69.1–75.2	71.3–75.6
30% IIV & 50% IOV	50.1–52.0	56.7–63.7	63.4–72.1	65.8–73.0
30% IIV & 60% IOV	50.0–50.3	54.2–61.9	61.7–68.5	62.5–71.4
F_1_ (%)				
30% IIV & 0% IOV	97.5–97.4	69.6–75.4	93.2–97.8	94.5–98.5
30% IIV & 10% IOV	83.8–97.1	69.7–68.1	82.6–85.9	83.3–87.4
30% IIV & 20% IOV	50.9–78.5	64.7–66.1	76.0–76.8	75.6–76.9
30% IIV & 30% IOV	19.3–59.2	56.3–67.3	68.0–75.1	69.0–75.7
30% IIV & 40% IOV	4.43–31.5	51.9–59.2	64.2–70.0	66.6–70.6
30% IIV & 50% IOV	0.60–12.7	44.5–55.4	57.7–66.8	60.6–67.5
30% IIV & 60% IOV	NC–2.55	39.3–53.8	55.0–62.6	56.2–66.4
MCC (%)				
30% IIV & 0% IOV	94.9–94.8	60.4–65.9	87.9–95.7	90.0–97.0
30% IIV & 10% IOV	70.6–94.1	59.1–59.0	73.7–77.7	74.5–79.6
30% IIV & 20% IOV	37.0–65.5	47.5–56.0	63.4–67.1	63.1–67.3
30% IIV & 30% IOV	14.1–46.6	30.1–53.0	48.3–63.7	50.4–65.1
30% IIV & 40% IOV	2.53–21.0	23.0–36.0	39.7–53.5	44.4–54.4
30% IIV & 50% IOV	2.24–8.91	14.8–29.4	27.8–46.6	32.6–48.9
30% IIV & 60% IOV	NC–3.02	9.60–25.4	24.4–38.8	26.1–44.8
κ (%)				
30% IIV & 0% IOV	94.8–94.7	53.4–60.5	87.2–95.6	89.5–97.0
30% IIV & 10% IOV	69.9–94.0	52.9–51.6	70.4–75.3	71.4–77.6
30% IIV & 20% IOV	30.2–62.9	43.5–48.8	59.9–62.3	59.4–62.4
30% IIV & 30% IOV	7.40–39.7	28.1–48.2	45.9–59.3	47.8–60.4
30% IIV & 40% IOV	0.70–14.0	21.4–33.5	38.2–50.3	42.6–51.2
30% IIV & 50% IOV	0.20–3.90	13.3–27.3	26.8–44.2	31.5–46.0
30% IIV & 60% IOV	0.00–0.60	8.40–23.8	23.3–36.9	25.0–42.8

Values represent the range calculated from simulated studies with 12 and 30 subjects. When statistics do not change between 12 and 30 subjects, unique values are presented instead of ranges. F_1_—harmonic mean of sensitivity and precision; κ—Cohen’s Kappa; MCC—Matthews’ correlation coefficient; NPV—negative predictive value. NC—not calculated.

## Data Availability

The data can be shared up on request.

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
