# Peer review of "Predictive Potential of Cmax Bioequivalence in Pilot Bioavailability/Bioequivalence Studies, through the Alternative ƒ2 Similarity Factor Method"

_pharmaceutics, 2023, doi:10.3390/pharmaceutics15102498_

Round 1
Reviewer 1 Report
The manuscript entitled “Predictive Potential of Cmax Bioequivalence in Pilot Bioavailability/Bioequivalence Studies, Through the Alternative ƒ2 Similarity Factor” evaluates the proposed ƒ2 factor in comparison with the standard average bioequivalence in more extreme scenarios, using a true GMR of 90% or 111% for truly bioequivalent formulations, and 80% or 125% for truly bioinequivalent formulations, in order to better drive conclusions on the potential of this analysis method. However, the presented article could be published in Pharmaceutics after revision for the following.
1. Introduction part should be enriched with a sufficient background as well as a justification for the current research purpose.
2. The discussion part is so week and need to discuss in detail with sufficient references.
3. Distribution of Observed Tmax should be placed in the original article.
4. Limitations of the study should be provided
Author Response
- Introduction part should be enriched with a sufficient background as well as a justification for the current research purpose.
Authors’ Response:
The authors acknowledge Reviewer’s comments.
This work is intended to be the continuation of an initial work already published by the authors1, in which an alternative ƒ2 factor applied to the comparison of the geometric mean pharmacokinetic profiles is proposed as a simple methodology to analyse pharmacokinetic data from pilot bioequivalence studies to overcome the sensitivity of such data to intra-subject variability.
In order to more appropriately contextualize this second work, as suggested by Reviewer, the Authors added the following paragraph between lines 37 and 43.
«… These pharmacokinetic clinical studies are designed to demonstrate comparable bioavailability or bioequivalence, defined as the absence of a significant difference in the rate and extent to which the active substance in pharmaceutical equivalents or pharmaceutical alternatives medicinal products becomes available at the site of drug action when administered at the same molar dose, under similar conditions. Claiming bioequivalence between two products assumes, an equivalent therapeutic efficacy and safety.»
Additionally, paragraph included between lines 44 and 46 was enriched with sentence:
«When companies are uncertain on the potential of a new formulation to be bioequivalent to a so called Reference product…»
Editorial revision and minor adjustments on sentences were also performed in paragraphs included between lines 47 and 70.
In order to more appropriately justify this second work, the authors have added the following sentence between lines 70 and 71, which it is thought to complete the paragraph and the reasoning for this second work:
«Considering that during drug product development less favourable GMR are commonly expected,…».
1 https://doi.org/10.3390/pharmaceutics15051430
- The discussion part is so week and need to discuss in detail with sufficient references.
Authors’ Response:
The authors acknowledge Reviewer’s comments.
Discussion section has been revised, as suggested by the reviewer.
- Distribution of Observed Tmax should be placed in the original article.
Authors’ Response:
The authors acknowledge Reviewer’s comments.
Distribution of tmax values is presented in supplementary materials, in table format and plots.
- Limitations of the study should be provided
Authors’ Response:
The authors acknowledge Reviewer’s comments.
The limitations of the alternative ƒ2 factor methodology, considering the simulations frame tested, is described between lines 496 and 499, with the following sentence:
«This decision tree is thought to assist companies on their decision to move forward with a full-size pivotal study, for drugs following a one compartment model, with median tmax ranging from 0.75 to 8 h, a mean elimination half-life of approximately 4.6 h and a mean volume of distribution of approximately 60 L, as the limits of tested scenarios.»
Reviewer 2 Report
The manuscript entitled “Predictive Potential of Cmax Bioequivalence in Pilot Bioavailability/Bioequivalence Studies, Through the Alternative ƒ2 Similarity Factor” is very interesting and very novel. I would recommend accepting for publication with minor modifications.
The major consideration – Gmean ƒ2 values and variability scenarios tested (20% – 60% IOV) will be applicable to any oral delivered product. How the process will be differ from API to API or process to process. Please explain.
Will be the process helpful for selection of population for NDA formulation as well?
Provide the references for Table 1.
Check the equation numbers in the manuscript.
Define 30% IIV & 0% IOV in supplementary tables.
Please define Gmean in the abstract. Use abbreviations uniformly (BE, etc).
Write the model examples as validation of the approach.
How the process will be effective with high ka value products
Author Response
- The manuscript entitled “Predictive Potential of Cmax Bioequivalence in Pilot Bioavailability/Bioequivalence Studies, Through the Alternative ƒ2 Similarity Factor” is very interesting and very novel. I would recommend accepting for publication with minor modifications.
Authors’ Response:
The authors acknowledge Reviewer’s comments.
- The major consideration – Gmean ƒ2 values and variability scenarios tested (20% – 60% IOV) will be applicable to any oral delivered product. How the process will be differ from API to API or process to process. Please explain.
Authors’ Response:
The authors acknowledge Reviewer’s comment.
The developed methodology is independent from API and/or manufacturing process. However, given that simulations are performed within limits, these constitute the applicable limitations of the method, which were included in the revision of the manuscript between lines 496 and 499, with the following sentence:
«This decision tree is thought to assist companies on their decision to move forward with a full-size pivotal study, for drugs following a one compartment model, with median tmax ranging from 0.75 to 8 h, a mean elimination half-life of approximately 4.6 h and a mean volume of distribution of approximately 60 L, as the limits of tested scenarios.»
- Will be the process helpful for selection of population for NDA formulation as well?
Authors’ Response:
The authors acknowledge Reviewer’s comment.
The developed methodology is applicable for comparative bioavailability or bioequivalence purposes, both in the development of generic medicinal products and innovators. For the last case, bioequivalence trials are performed to link commercial formulations to clinical trials formulations or in the case of post approval changes.
- Provide the references for Table 1. Check the equation numbers in the manuscript.
Authors’ Response:
The authors acknowledge Reviewer’s comments.
The Materials and Methods section was reduced due to similarity to the previous article (Henriques et al. 2023).
Hence, Table 1 was removed, and the explanation of the performance measurement was reduced to a brief explanation of the metrics, that is referenced by the previous article (Henriques et al. 2023), and two other articles.
Equations numbers were reviewed.
- Define 30% IIV & 0% IOV in supplementary tables.
Authors’ Response:
Supplementary material was updated to include the definition of IIV and IOV.
- Please define Gmean in the abstract. Use abbreviations uniformly (BE, etc).
Authors’ Response:
The authors acknowledge Reviewer’s notice.
Authors have defined the Gmean abbreviation in the abstract.
- Write the model examples as validation of the approach.
Authors’ Response:
The authors acknowledge Reviewer’s comment.
Model outcomes are presented in the supplementary materials.
- How the process will be effective with high ka value products
Authors’ Response:
The authors acknowledge Reviewer’s comment.
Given that simulations are performed within limits, these constitute the applicable limitations of the method. In this work ka values ranged from 0.484 to 1.22 h-1, which combined in a one compartment model with ke value of 0.15 h-1 and a volume of distribution of 60 L, originate median tmax values in the range of 0.75 and 8 h.
These limitations of the method were included in the revision of the manuscript between lines 496 and 499, with the following sentence:
«This decision tree is thought to assist companies on their decision to move forward with a full-size pivotal study, for drugs following a one compartment model, with median tmax ranging from 0.75 to 8 h, a mean elimination half-life of approximately 4.6 h and a mean volume of distribution of approximately 60 L, as the limits of tested scenarios.»
The authors are also working in another manuscript on which this alternative methodology is being applied using results from positive and negative pivotal bioequivalence trials, as a validating procedure for the methodology. These bioequivalence results are not included in the simulations frame. For the moment it can be disclosed that this methodology seems to be promising.
Reviewer 3 Report
pharmaceutics-2614531
Predictive Potential of Cmax Bioequivalence in Pilot Bioavailability/Bioequivalence Studies, Through the Alternative ƒ2 Similarity Factor
The manuscript by Henriques et al. is a follow-up study to a previous one. The authors evaluated ƒ2 factor in comparison with the standard average bioequivalence. Overall, the manuscript was well prepared, and the data were sufficient for the conclusion. Below are some suggestions to improve this manuscript.
1. The Tables are in different formats. Please make them consistent and follow the journal guidelines.
2. Many parts of the methods repeated the authors’ previous article (ref. #3). Therefore, the authors may cite the article here and omit these parts.
3. Please check and correct some typos and grammar errors.
Minor editing of English language required
Author Response
The manuscript by Henriques et al. is a follow-up study to a previous one. The authors evaluated ƒ2 factor in comparison with the standard average bioequivalence. Overall, the manuscript was well prepared, and the data were sufficient for the conclusion. Below are some suggestions to improve this manuscript.
Authors’ Response:
The authors acknowledge Reviewer’s comments.
- The Tables are in different formats. Please make them consistent and follow the journal guidelines.
Authors’ Response:
The authors acknowledge Reviewer’s comments.
In fact, Table 1 was not in the standard table format required by the journal. This table was removed in order to simplify the Materials and Methods section.
- Many parts of the methods repeated the authors’ previous article (ref. #3). Therefore, the authors may cite the article here and omit these parts.
Authors’ Response:
The authors acknowledge Reviewer’s comments.
The Materials and Methods section was reduced due to similarity to the previous article (Henriques et al. 2023).
- Please check and correct some typos and grammar errors.
Authors’ Response:
The authors acknowledge Reviewer’s comments.
The article was reviewed, and typos and grammar errors corrected.
Round 2
Reviewer 1 Report
The manuscript was modified and could me accepted in its present form